# Synthesis, Characterization, and CO_2_/N_2_ Separation Performance of POEM-*g*-PAcAm Comb Copolymer Membranes

**DOI:** 10.3390/polym13020177

**Published:** 2021-01-06

**Authors:** Byeong Ju Park, Na Un Kim, Chang Soo Lee, Jong Hak Kim

**Affiliations:** Department of Chemical and Biomolecular Engineering, Yonsei University, 50 Yonsei-ro, Seodaemun-gu, Seoul 03722, Korea; blackholebob@naver.com (B.J.P.); ian4045@naver.com (N.U.K.); ch1ch2ch3@naver.com (C.S.L.)

**Keywords:** comb copolymer, microphase-separation, polymer membrane, CO_2_ separation

## Abstract

Alcohol-soluble comb copolymers were synthesized from rubbery poly(oxyethylene methacrylate) (POEM) and glassy polyacrylamide (PAcAm) via economical and facile free-radical polymerization. The synthesis of comb copolymers was confirmed by Fourier-transform infrared and proton nuclear magnetic resonance spectroscopic studies. The bicontinuous microphase-separated morphology and amorphous structure of comb copolymers were confirmed by wide-angle X-ray scattering, differential scanning calorimetry, and transmission electron microscopy. With increasing POEM content in the comb copolymer, both CO_2_ permeability and CO_2_/N_2_ selectivity gradually increased. A mechanically strong free-standing membrane was obtained at a POEM:PAcAm ratio of 70:30 wt%, in which the CO_2_ permeability and CO_2_/N_2_ selectivity reached 261.7 Barrer (1 Barrer = 10^−10^ cm^3^ (STP) cm cm^−2^ s^−1^ cmHg^−1^) and 44, respectively. These values are greater than those of commercially available Pebax and among the highest separation performances reported previously for alcohol-soluble, all-polymeric membranes without porous additives. The high performances were attributed to an effective CO_2_-philic pathway for the ethylene oxide group in the rubbery POEM segments and prevention of the N_2_ permeability by glassy PAcAm chains.

## 1. Introduction

Growing concerns about global warming resulting from accelerated industrialization has led to the increasing demand for advanced gas purification and CO_2_ separation technologies [1]. Compared to other gas separation technologies, such as adsorption, absorption, and cryogenics, membrane technology has many advantages, including high energy efficiency, low operating costs, small footprint, and easy scale-up [2,3,4,5,6,7]. Polymers are attractive materials for preparing membranes because of their diversity, simple manufacturing methods, good processability, and high separation performance [8]. Furthermore, it is possible to fabricate various membranes for different applications owing to the diverse monomers available and the polymers that are synthesized from them with different structural properties. In recent years, membrane technology for CO_2_ separation has attracted significant research interest [9,10,11,12]. In particular, studies have reported on CO_2_/N_2_, CO_2_/CH_4_, and CO_2_/H_2_ separation techniques [13,14,15,16,17]. However, conventional polymeric membranes suffer from a tradeoff between permeability and selectivity [18,19]; high selectivity is usually accompanied by low permeability and vice versa. Therefore, it is important to develop innovative membranes that show high permeability and high selectivity simultaneously.

Gas molecules permeate through a polymeric membrane via a solution-diffusion mechanism, in which the permeability of the gas is expressed as the product of its solubility and diffusivity [20]. Thus, an improvement in the solubility or diffusivity leads to increased permeability. Because diffusivity is a kinetic factor, it is mainly affected by the kinetic diameter of the permeant gas and the free volume of the polymeric membrane [21,22]. On the other hand, solubility is a thermodynamic factor, which is related to the interactions between gas molecules and the polymer [23,24]. It is well known that the presence of polar groups, such as ether oxygen (–C–O–C–) in polymer chains, results in high CO_2_ solubility because of their strong affinity toward CO_2_ gas molecules due to dipole–quadrupole interactions [25,26,27].

Poly(ethylene oxide) (PEO) is a representative rubbery polymer containing polar ether groups, which can enhance CO_2_ solubility based on the Lewis acid–base interactions between the CO_2_ molecule (acid) and the ether oxygen (base) [28,29]. However, it is difficult to prepare gas separation membranes using high-molecular-weight PEO owing to their high crystallinity, which leads to structural defects and low permeability [8]. To overcome this problem, PEO-based block copolymers have been investigated, such as poly(amide-b-ethylene oxide) (PA-b-PEO, well known as Pebax), which is covalently bonded to different polymer chains at a single point [30,31,32,33]. However, the synthesis of block copolymers is sensitive to impurities, resulting in high processing costs [34]. Comb copolymers have been considered as alternatives to block copolymers because of their easy synthesis and low cost. Poly(oxyethylene methacrylate) (POEM), analogous to PEO with an amorphous nature, has been investigated [35,36,37]. However, POEM has liquid-like properties and poor mechanical strength, which limits its application to gas separation membranes.

In this study, we report the synthesis of poly(oxyethylene methacrylate)-g-poly(acrylamide) (POEM-*g*-PAcAm) comb copolymer via facile free-radical polymerization for application as a CO_2_ separation membrane. The POEM content was varied from 50% to 100% to optimize the CO_2_/N_2_ gas separation performance. We expected that the POEM segments would form an efficient CO_2_-philic pathway to enhance the solubility of the CO_2_ gas molecules, while the rigid poly(acrylamide) (PAcAm) segments would compensate for the liquid-like properties of the POEM chains. The POEM-*g*-PAcAm free-standing membranes were fabricated by the solution casting method, and they were investigated by Fourier-transform infrared (FT–IR) spectroscopy and proton nuclear magnetic resonance (^1^H NMR) spectroscopy. The structural and thermal properties of the comb copolymers were characterized by X-ray diffraction (XRD) analysis, differential scanning calorimetry (DSC), and transmission electron microscopy (TEM). The CO_2_/N_2_ gas separation performances were analyzed using a time-lag method at 35 °C.

## 2. Materials and Methods

### 2.1. Materials

Monomers for polymerization such as acrylamide (AcAm, M_n_ = 71.08 g/mol) and poly(oxyethylene methacrylate) (POEM, M_n_ = 500 g/mol) were purchased from Sigma-Aldrich (St. Louis, MO, USA). The initiator of the free radical polymerization, 2,2-azobis(2-methylpropionitrile) (AIBN, 98%), was obtained from Acros Organics. *N*,*N*-Dimethylformamide (DMF), isopropyl alcohol (IPA), hexane (n-hexane, 95%), and absolute ethanol were purchased from J. T. Baker (Phillipsburg, NJ, USA). All solvents and chemicals were of reagent grade and were used without further purification.

### 2.2. Synthesis of POEM-g-PAcAm Comb Copolymers

A series of POEM-g-PAcAm copolymers were synthesized by free radical polymerization. The total amount of the POEM macromonomer and AcAm monomers was fixed at 10 g. First, different amounts of POEM and AcAm were dissolved in DMF (50 mL) in a round-bottom flask. Subsequently, 0.002 g of AIBN was added to the polymer solution to initiate polymerization. After N_2_ purging, the mixture was heated to 70 °C for 18 h. The polymer solution was then precipitated using an excessive amount of a mixture of IPA and n-hexane (3:7). The precipitation process was repeated thrice to remove unreacted monomers, and the obtained polymer was dried in a vacuum oven overnight at 25 °C. Various weight ratios of POEM:AcAm were used, including 1:0, 7:3, 6:4, 5:5, and 0:1, which were referred to as PPOEM, PAA73, PAA64, PAA55, and PAcAm, respectively.

### 2.3. Preparation of POEM-g-PAcAm Membranes

The POEM-g-PAcAm free-standing membranes were prepared via a solution casting method. A 10 wt% POEM-g-PAcAm solution was prepared by dissolving the copolymer in pure ethanol at room temperature. The prepared solutions were cast onto circular glass dishes and slowly dried at room temperature for three days. The membranes were further dried at 50 °C in a vacuum oven to completely eliminate any residual solvents to obtain the free-standing POEM-g-PAcAm membranes with a thickness of 70–80 μm. Pure polyPOEM (POEM homopolymer) membrane could not be prepared because of its poor mechanical strength.

### 2.4. Characterization

FTIR spectra of the synthesized polymers were obtained in the frequency range 4000–600 cm^−1^. Polymerization was further confirmed by ^1^H NMR analysis. The POEM-g-PAcAm comb copolymers were dissolved in deuterium oxide (D_2_O), and the ^1^H NMR spectra were obtained using a 400-MHz FT-BNR spectrometer (ADVANCE III HD 400, Bruker Biospin, North Billerica, MA, USA. A field-emission scanning electron microscope (FE-SEM, JSM-7001F, JEOL Ltd., Tokyo, Japan) was used to characterize the surface and cross-sectional morphology of the polymeric membranes. For high-resolution transmission electron microscopy (HR-TEM, JEM-3010, JEOL Ltd., Tokyo, Japan), 0.5 wt% polymer solution in ethanol was directly cast onto a TEM grid and dried in a vacuum oven at room temperature for 1 day to completely remove the solvent. Wide-angle X-ray scattering (WAXS) analysis was carried out using a Rigaku 18 kW rotating anode X-ray generator with Cu-Kα radiation (λ = 1.5405 nm) operated at 40 kV and 300 mA. The thermal behavior of the copolymers was analyzed using a differential scanning calorimeter (DSC8000, Perkin Elmer, Waltham, MA, USA) operated at a heating rate of 10 °C/min in air. The polymer sample was first heated from –70 °C to 100 °C, then cooled to −70 °C, and again heated from −70 °C to 100 °C. The second scanning data were used to determine the thermal transition of the copolymer. 

### 2.5. Gas Permeation Measurement

The pure gas permeation properties of the membranes were investigated by a time-lag method using a constant volume/variable pressure apparatus (Airrane Co. Ltd., Cheongju, Korea), according to a previously reported procedure. The downstream pressure was maintained at less than 2 Torr, which was much lower than the upstream pressure (760 Torr). The gas permeability was calculated from the steady-state rate of increase in the downstream pressure while the volume was kept constant. Five replicates of each membrane were tested for reproducibility, and the average error was approximately ± 5%. The gas permeability (P) was calculated using the following equation:(1)$$P=\frac{1}{760}\cdot \frac{V}{A}\cdot \frac{273.15}{T}\cdot \frac{l}{∆p}\cdot \frac{\mathrm{dp}}{\mathrm{dt}}$$
where P is the gas permeability in Barrer (1 Barrer = 10^−10^ cm^3^ (STP) cm cm^−2^ s^−1^ cm Hg^−1^), A is the effective membrane area (cm^2^), V is the volume of the chamber (cm^3^), T is the experimental temperature (K), p is the transmembrane pressure difference (cmHg), l is the membrane thickness (cm), and dp/dt is the steady-state rate of pressure rise (mmHg/s) on the downstream side. The CO_2_/N_2_ selectivity (α) of the membrane was calculated using the ratio of the permeabilities of the two pure gases under the same conditions:(2)$$\alpha_{CO_{2}/N_{2}}=\;P_{CO_{2}}/P_{N_{2}}$$

## 3. Results and Discussion 

### 3.1. Synthesis of POEM-g-PAcAm Copolymer 

The copolymerization of POEM and AcAm is illustrated in Scheme 1. A series of POEM-g-PAcAm comb copolymers with different POEM to AcAm ratios were synthesized via facile one-pot free radical polymerization. The physicochemical properties of the POEM-g-PAcAm comb copolymer stem from the combination of the flexible hydrophilic POEM and rigid PAcAm chains. The rigid PAcAm segments in the comb copolymer compensate for the liquid-like properties of the POEM chains and endow the copolymer with good mechanical stability. The ethylene oxide segments of POEM with high chain mobility could improve the CO_2_ affinity of the membrane and also prevent excessive densification of the polymer matrix. Additionally, PAcAm chains contain secondary amine groups, which possess high capacity for CO_2_ loading due to specific interactions between the basic amines and acidic CO_2_.

The physical properties of the POEM-g-PAcAm comb copolymers were highly dependent on their composition. As shown in Figure 1a, a rigid solid state was observed for PAA73, PAA64, and PAA55, while PPOEM showed highly viscous liquid-like properties. These results indicate that the solid-like properties of the copolymers increased with increasing acrylamide content. The rigidity of the PAcAm segments can enhance the mechanical properties of the copolymers and help to form free-standing membranes. Pictures depicting the good mechanical strength of the POEM-g-PAcAm (PAA73) membrane are shown in Figure 1b,c. When the membrane was stretched to both sides, the horizontal length of the membrane increased over two times while maintaining the membrane form, indicating good mechanical strength.

The successful polymerization of the POEM-g-PAcAm comb copolymers was confirmed by FTIR and ^1^H NMR spectroscopy, as shown in Figure 2. The FTIR spectra of the two monomers (POEM, AcAm) and the POEM-g-PAcAm comb copolymer are shown in Figure 2a. For the AcAm monomer, two absorption bands at 3340 and 1425 cm^−1^ were observed, which are assigned to the –NH_2_ stretching and C–N stretching vibrations, respectively. Strong bands at 2866 and 1717 cm^−1^, attributed to the stretching vibrations of –CH_3_ and C=O groups, respectively, were observed in the POEM macromonomer. A sharp band at 1662 cm^−1^ was observed in the AcAm monomer and POEM-g-PAcAm comb copolymer owing to the stretching vibration mode of C=O in acrylamide [38]. In the POEM-g-PAcAm comb copolymers, characteristic absorption bands of both AcAm and POEM monomers were observed, indicating that POEM-g-PAcAm comb copolymers were synthesized successfully. The ^1^H NMR spectra of the POEM-g-PAcAm comb copolymer is shown in Figure 2b. The composition of the comb copolymers was calculated by integrating each of the corresponding chemical shifts, i.e., –CH_2_– (c) and OCH_3_ (d) protons of POEM at 3.6 and 3.3 ppm, respectively [37,39], and –CH– (b) and –NH_2_– (e) protons of acryl amide at 1.6 and 7.8 ppm, respectively [40,41]. As shown in Table 1, the compositions of the synthesized copolymers were consistent with the feed ratio of the monomers used, indicating a controlled polymerization reaction.

### 3.2. Structural and Morphological Properties

DSC analysis was carried out to investigate the glass transition temperature (T_g_) of the POEM-g-PAcAm comb copolymers (Figure 3a). No endothermic crystalline melting peaks were observed for any of the polymers, indicating their amorphous nature. All the POEM-g-PAcAm comb copolymers had similar T_g_ values at approximately −55 °C, which was attributed to the segmental motion of the rubbery POEM chains [42]. The weak T_g_ value observed at approximately 134 °C was attributed to the motion of the glassy PAcAm chain [43]. The two distinct T_g_ values demonstrate that POEM-g-PAcAm comb copolymers have microphase-separated nanostructures with rubbery POEM as well as glassy PAcAm characteristics.

The structures of the homopolymers and comb copolymers with various ratios were investigated using WAXS analysis. Figure 3b shows the WAXS curves of pure PPOEM (POEM homopolymer), PAA73, PAA64, PAA55, and PAcAm (acrylamide homopolymer). No sharp crystalline peaks were observed for any of the polymers, indicating a completely amorphous nature, which is in good agreement with the DSC analysis. Peaks centered at 2θ = 20.8° in PAcAm and at 2θ = 20.2° in PPOEM were observed [44,45,46]. Using Bragg’s law (2d sin θ = *n**λ*), Law (2dsinθ=nλ), the d-spacing values were determined to be 4.5 and 4.4 Å for PAcAm and PPOEM, respectively.

The microstructures of the POEM-g-PAcAm comb copolymer were characterized by TEM (Figure 4). In TEM images, domains with higher electron density appear dark, while those with lower electron density appear bright. In the PAA55 membrane, dark, isolated POEM domains were uniformly dispersed in the bright PAcAm matrix (Figure 4a). With increasing amount of POEM segments, the area of isolated POEM regions increased (Figure 4b,c). Bicontinuous microphase-separated structures were observed in PAA73 and PAA64 copolymers, which is consistent with the two T_g_ values observed in the DSC analysis. To confirm the distribution of the nitrogen elements (from PAcAm segments) across the membrane, EDS spectra and scanning transmission electron microscopy (STEM) of the membrane were collected and are shown in Figure 5. A large amount of nitrogen was observed in the EDS image of PAA55, which gradually decreased with the increase in POEM content in the copolymer. In both PAA73 and PAA64, both domains (dark POEM and yellow PAcAm for EDS images; bright POEM and dark PAcAm for STEM images) were observed to be bicontinuous and microphase-separated, which is consistent with the TEM images above. This morphology may provide an effective pathway for efficient CO_2_ gas transport through the membranes.

### 3.3. CO_2_/N_2_ Separation Performance

Figure 6 and Table 2 show the CO_2_/N_2_ separation performance of the POEM-g-PAcAm comb copolymer membranes using the time-lag method at 35 °C and 760 Torr (1 bar). As mentioned earlier, the permeability of gas molecules through a polymeric membrane is determined based on solubility and diffusivity. Because CO_2_ is more condensed than N_2_ (critical temperatures of 195 and 71 K, respectively), the solubility of CO_2_ is higher than that of N_2_. Furthermore, the kinetic diameter of CO_2_ (3.30 Å) is smaller than that of N_2_ (3.64 Å), which means that the diffusivity of CO_2_ gas molecules is always higher [47]. Therefore, the permeability of CO_2_ is higher than that of N_2_ because of the dual effect of higher solubility and higher diffusivity. With increasing POEM content, both the CO_2_ and N_2_ permeabilities increased gradually. However, the increase in the permeability of CO_2_ was significantly greater than that of N_2_. Thus, the CO_2_ permeability and CO_2_/N_2_ selectivity increased simultaneously because the CO_2_−philic segments in the POEM chains enhance the CO_2_ solubility of the membrane and form a microphase-separated channel for CO_2_ transport, as confirmed by TEM, EDS, and STEM analyses [48]. The bicontinuous microphase-separated structure was attributed to the selective interactions between POEM segments, which have much longer chain lengths compared to PAcAm segments, resulting in interconnected CO_2_-philic pathways in the membranes. These pathways increase the CO_2_ permeability significantly, while only slightly increasing the N_2_ permeability, thus improving the CO_2_/N_2_ selectivity, as illustrated in Figure 7. To clearly investigate the effect of the CO_2_-philic segments on the separation properties, the diffusivity and solubility values of POEM-g-PAcAm membranes were determined as shown in Table 3. The CO_2_ solubility of the membranes increased dramatically as the POEM content increased, indicating that the POEM chains have an excellent CO_2_-philicity.

Figure 8 shows a Robeson plot showing the tradeoff relationship between the selectivity and permeability of the polymeric membranes [18]. Among the POEM-g-PAcAm membranes, the PAA73 exhibited outstanding CO_2_/N_2_ separation performance on the Robeson upper bound and was superior to commercial Pebax block copolymer and other alcohol-soluble, all-polymeric membranes based on PEO or PEG polymer matrix without porous additives, such as metal organic frameworks (MOFs). The PAA73 showed the CO_2_ permeability of 261.7 Barrer and CO_2_/N_2_ selectivity of 44.0. The good solubility of the POEM-g-PAcAm comb copolymer in alcohol is a vital factor for commercial applications. The comb copolymer could be used as a selective coating layer for thin-film composite membranes because most of the porous polymer supports (e.g., polysulfone) do not dissolve in alcohol. Further studies can be carried out on the fabrication of thin-film composite membranes to improve the permeance of membranes in gas permeance unit (GPU).

## 4. Conclusions

In this study, high-performance gas separation membranes were prepared based on CO_2_-philic POEM-g-PAcAm comb copolymers consisting of rigid PAcAm segments and rubbery hydrophilic POEM segments. The successful synthesis of POEM-g-PAcAm comb copolymers was confirmed by FTIR and ^1^H NMR spectroscopy. The POEM-g-PAcAm comb copolymers have an amorphous structure as investigated by DSC and XRD analysis. The bicontinuous microphase-separated channel for CO_2_ transport was confirmed by TEM, EDS, and STEM analyses. The CO_2_-philic ethylene oxide (C–O–C) groups in the POEM chains of the copolymer improved the CO_2_ solubility of the membrane. The rigid PAcAm chains played a crucial role in minimizing the N_2_ permeability to enhance the selectivity. For the PAA73 membrane, the CO_2_ permeability was 261.7 Barrer and the CO_2_/N_2_ selectivity reached 44.0, which is among the highest separation performances previously reported for alcohol-soluble, all-polymeric membranes without porous additives. Furthermore, the POEM-g-PAcAm comb copolymers were synthesized via free radical polymerization, which is more economical than the synthesis of conventional block copolymers. The results reported herein suggest that the POEM-*g*-PAcAm comb copolymers have great potential for CO_2_ capture applications because of their simple fabrication process and high gas separation performance.

## Data Availability

Not applicable.

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
