# Peer review of "Synthesis, Characterization, and CO2/N2 Separation Performance of POEM-g-PAcAm Comb Copolymer Membranes"

_polymers, 2021, doi:10.3390/polym13020177_

Round 1

Reviewer 1 Report

1). 2.2. Synthesis of POEM-g-PAcAm comb copolymers

Page 2, lines 88-90…The precipitation process was repeated thrice to remove unreacted monomers, and the obtained polymer was dried in a vacuum oven overnight at ambient conditions.

…“at ambient conditions” – this means that drying was carried out at room temperature and atmospheric pressure. Then what is a vacuum oven for?

2)….3.2. Structural and morphological properties

Page 7. line 211 -    PAA63….. – error, replace with PAA64

3) … section 3.3 is missing

Author Response

Reviewer #1

Comments and Suggestions for Authors

1). 2.2. Synthesis of POEM-g-PAcAm comb copolymers

Page 2, lines 88-90…The precipitation process was repeated thrice to remove unreacted monomers, and the obtained polymer was dried in a vacuum oven overnight at ambient conditions.

…“at ambient conditions” – this means that drying was carried out at room temperature and atmospheric pressure. Then what is a vacuum oven for?

Response

In order to clarify, the words “at ambient conditions”, was replaced by “at 25 oC”. It means that the obtained polymer was dried in a vacuum oven at 25 oC to remove the residual solvents.

2)….3.2. Structural and morphological properties

Page 7. line 211 -    PAA63….. – error, replace with PAA64

Response

As you pointed out, there was an error on page 7. line 211. We have corrected it. Thank you very much for careful comment.

3) … section 3.3 is missing

Response

The section [3.3. CO2/N2 separation performance] is already included.

Reviewer 2 Report

In the submitted manuscript (ID: polymers-1035354), a polymer with the advantages of the unique properties of oxyethylene methacrylate and acrylamide monomers were prepared through polymerization, and applied in the selective separation of CO2/N2. Some interesting results are obtained, with extensive discussions and proper design. Overall, this manuscript can be considered for publication in the journal Polymers after clarification of following issues.

1) Regarding the analysis for the FTIR spectra, the illustration of the C=C bond is missing, by which the residue of monomer and the polymerization of the polymer can be indicated.

2) Figure 2b needs to be further adjusted, in which the peaks are overlap mutually and hard to distinguish.

3) The different ability for adsorbing CO2 and N2 needs to be further illustrated by N2 and CO2 adsorption-desorption test, and the pore size of membrane needs discussion as well.

4) The excellent mechanical strength and the vast pressure difference between two sides of the membrane are attracted our attention. Therefore, whether the vast pressure difference would make the membrane deformed in the gas separation test. If the membrane deforms in the separation, the deformation of pore size is the main factor resulting in the high selectivity for the CO2.

5) There are some grammatical mistakes in the text. Please check and impove the manuscript thoroughly.

Author Response

Reviewer #2

the submitted manuscript (ID: polymers-1035354), a polymer with the advantages of the unique properties of oxyethylene methacrylate and acrylamide monomers were prepared through polymerization, and applied in the selective separation of CO2/N2. Some interesting results are obtained, with extensive discussions and proper design. Overall, this manuscript can be considered for publication in the journal Polymers after clarification of following issues.

1) Regarding the analysis for the FTIR spectra, the illustration of the C=C bond is missing, by which the residue of monomer and the polymerization of the polymer can be indicated.

Response

As you pointed out, the absorption band of C=C bond at 1634 cm-1 has been marked in Figure (a). However, it is difficult to confirm that there is the residue of monomer left only with this because this band is overlapped with other bands.

2) Figure 2b needs to be further adjusted, in which the peaks are overlap mutually and hard to distinguish.

Response

As you suggested, the spacing of graphs in Figure 2b was adjusted to facilitate distinction.

3) The different ability for adsorbing CO2 and N2 needs to be further illustrated by N2 and CO2 adsorption-desorption test, and the pore size of membrane needs discussion as well.

Response

As you pointed out, the diffusivity and solubility of CO2 gas has been inserted in Table 3 additionally. We have added the following descriptions on page 8, line 236, “To clearly investigate the effect of the CO2-philic segments on the separation properties, the diffusivity and solubility values of POEM-g-PAcAm membranes were determined as shown in Table 3. The CO2 solubility of the membranes increased dramatically as the POEM content increased, indicating that the POEM chains have an excellent CO2-philicity.”

Regarding the pore size of membrane, all POEM-g-PAcAm membranes were free-standing membrane with dense structure. Therefore, these polymer membranes do not have specific pores. When we measured the free volume of membrane using XRD results and all POEM-g-PAcAm membranes showed 4.4 â„«, as described on page 7.

4) The excellent mechanical strength and the vast pressure difference between two sides of the membrane are attracted our attention. Therefore, whether the vast pressure difference would make the membrane deformed in the gas separation test. If the membrane deforms in the separation, the deformation of pore size is the main factor resulting in the high selectivity for the CO2.

Response

All POEM-g-PAcAm membranes are rubbery free-standing membrane without pores inside and thus there was no significant effect of pressure up to 5 bar on the membrane structure and selectivity.

5) There are some grammatical mistakes in the text. Please check and improve the manuscript thoroughly.

Response

As you pointed out, I have double-checked the manuscript thoroughly and corrected. Our manuscript has been proofread by Editage (http://www.editage.co.kr, http://www.cactusglobal.com)

Reviewer 3 Report

This paper describes synthesis and application of POEM-g-PAcAm comb copolymer.  The authors succeeded in the development of polymer materials for CO2/N2 separation. The findings in this paper are of basic interest and the copolymer materials are the substances of considerable interest for wide fields of chemistry, I think. Therefore, I recommend publication in Polymers.after correction of the following point.

1) Page 4, line 135; The copolymerization scheme of POEM-g-PAcAm is illustrated in Scheme 1. - -> The copolymerization of POEM and g-PAcAm is illustrated in Scheme 1.

Author Response

Reviewer #3

This paper describes synthesis and application of POEM-g-PAcAm comb copolymer. The authors succeeded in the development of polymer materials for CO2/N2 separation. The findings in this paper are of basic interest and the copolymer materials are the substances of considerable interest for wide fields of chemistry, I think. Therefore, I recommend publication in Polymers.after correction of the following point.

1) Page 4, line 135; The copolymerization scheme of POEM-g-PAcAm is illustrated in Scheme 1. - -> The copolymerization of POEM and g-PAcAm is illustrated in Scheme 1

Response

As suggested, we have modified it.